# A conserved regulatory program initiates lateral plate mesoderm emergence across chordates

Karin D. Prummel[1,11], Christopher Hess[1,11], Susan Nieuwenhuize[1], Hugo J. Parker [2,3], Katherine W. Rogers[4], Iryna Kozmikova[5], Claudia Racioppi[6], Eline C. Brombacher[1], Anna Czarkwiani [7], Dunja Knapp [7], Sibylle Burger[1], Elena Chiavacci[1], Gopi Shah [8], Alexa Burger[1], Jan Huisken [8,9], Maximina H. Yun [7,8], Lionel Christiaen [6], Zbynek Kozmik[5], Patrick Müller [4], Marianne Bronner[10], Robb Krumlauf[2,3] & Christian Mosimann [1]

Cardiovascular lineages develop together with kidney, smooth muscle, and limb connective tissue progenitors from the lateral plate mesoderm (LPM). How the LPM initially emerges and how its downstream fates are molecularly interconnected remain unknown. Here, we isolate a pan-LPM enhancer in the zebrafish-specific *draculin* (*drl*) gene that provides specific LPM reporter activity from early gastrulation. *In toto* live imaging and lineage tracing of *drl*-based reporters captures the dynamic LPM emergence as lineage-restricted mesendoderm field. The *drl* pan-LPM enhancer responds to the transcription factors EomesoderminA, FoxH1, and MixL1 that combined with Smad activity drive LPM emergence. We uncover specific activity of zebrafish-derived *drl* reporters in LPM-corresponding territories of several chordates including chicken, axolotl, lamprey, *Ciona*, and amphioxus, revealing a universal upstream LPM program. Altogether, our work provides a mechanistic framework for LPM emergence as defined progenitor field, possibly representing an ancient mesodermal cell state that predates the primordial vertebrate embryo.

[1] Institute of Molecular Life Sciences, University of Zurich, Zürich 8057, Switzerland. [2] Department of Anatomy and Cell Biology, Kansas University Medical Center, Kansas City, KS 66160, USA. [3] Stowers Institute for Medical Research, Kansas City, MO 64110, USA. [4] Friedrich Miescher Laboratory of the Max Planck Society, Tübingen 72076, Germany. [5] Institute of Molecular Genetics of the ASCR, Prague 142 20, Czech Republic. [6] Center for Developmental Genetics, Department of Biology, New York University, New York, NY 10003, USA. [7] TUD-CRTD Center for Regenerative Therapies Dresden, Dresden 01307, Germany. [8] Max Planck Institute of Molecular Cell Biology and Genetics, Dresden 01307, Germany. [9] Morgridge Institute for Research, Madison, WI 53715, USA. [10] Division of Biology and Biological Engineering, California Institute of Technology, Pasadena, CA 91125, USA. [11] These authors contributed equally: Karin D. Prummel, Christopher Hess. Correspondence and requests for materials should be addressed to C.M. (email: christian.mosimann@imls.uzh.ch)

Key cell fates and organ systems in vertebrates emerge from multipotent progenitors within the embryonic mesoderm. Following gastrulation, the vertebrate mesoderm has been classically described to partition into axial, paraxial, and ventro-lateral domains[1]. The latter, referred to as lateral plate mesoderm (LPM), is composed of highly motile cells and is mainly defined by its position adjacent to the somite-forming paraxial mesoderm. Transplantation and lineage tracing experiments in several species have established that the LPM contains progenitors of the circulatory system, smooth muscles, the kidneys (in amniotes often demarcated as intermediate mesoderm), and the limb connective tissue anlagen[2–4]. During segmentation, the LPM principally segregates into the anterior LPM (ALPM) and posterior LPM (PLPM), which further divides into dorsal and ventral domains (somatopleure and splanchnopleure, respectively). Several transcription factors including Hand1/2, Tbx5, Osr1, FoxF1, Prrx1, Mesp1, and Etv2 are expressed in LPM territories and play overlapping roles in cell fate determination[2,3,5], albeit not always with an evolutionarily conserved function[6]. It remains incompletely understood how the LPM arises from an initial mesodermal population that goes on to form distinct endodermal and mesodermal progenitors. This is partly due to the lack of tools and markers to track LPM emergence genetically during development. Further, whether the LPM initially emerges as morphogenetic field in a molecularly coherent unit or as a loosely connected assembly of progenitor cells remains unclear.

Assessing the evolutionary context by which the LPM emerged as a developmental entity also remains challenging, in particular in extant jawless vertebrates such as lamprey or chordate models that do not form the full spectrum of LPM derivatives. Ancestral gene-regulatory repertoires that control higher-order structures in vertebrates previously have been indicated for somatic muscle in lamprey[7] or for the putative equivalents of cardiac and hematopoietic progenitors in amphioxus[8]. Anterior-to-posterior expression domains of key LPM transcription factors including Tbx1/10 and Hand are conserved in lampreys and amphioxus[9]. Furthermore, the tunicate *Ciona* forms cardiac lineages that display genetic regulatory circuits homologous to the cardiac LPM progenitors found in vertebrates[10]. These observations suggest the existence of an ancient regulatory program that delineated prospective LPM progenitors in a common chordate ancestor, dating back to the Cambrian explosion 520–540 million years ago.

Several mammalian *cis*-regulatory elements with broad LPM activity have been reported; these include an upstream enhancer of mouse and human *HoxB6*[11], an upstream enhancer of mouse *Gata4*[12], and a downstream enhancer of mouse *Bmp4*[13]. In line with a ventral LPM origin, the *Gata4* LPM enhancer responds to Smads downstream of BMP signaling[12]. Nonetheless, the activities driven by these enhancer elements in mice confine to the PLPM and are seemingly not pan-LPM readouts. In zebrafish, the ventrally and marginally emerging LPM forms during somitogenesis into a patchwork of bilateral gene expression domains, including of the conserved LPM genes *hand2*, *pax2.1*, *scl*, *lmo2*, *etv2*, and *tbx5*[5]. In contrast, transgenic reporters based on the 6.35 kb *cis*-regulatory region of the zebrafish-specific gene *draculin* (*drl*) selectively label the entire LPM from its emergence during gastrulation through initial differentiation[14]. Cre/*lox*-mediated genetic lineage analysis has established that early *drl* reporter expression in zebrafish labels the LPM progenitors forming cardiovascular, blood, kidney, intestinal smooth muscles (iSMCs), and pectoral fin mesenchyme fates[14–16]. While *drl* as putative multimer zinc-finger gene has no obvious ortholog in other vertebrates[14,17,18], these observations suggest that the 6.35 kb *drl* region harbors *cis*-regulatory elements active throughout the prospective LPM starting from gastrulation, raising the

possibility that these regulatory elements read out a pan-LPM program.

Here, we dissect the 6.35 kb *drl cis*-regulatory elements and uncover an intronic enhancer, *+2.0drl*, that in zebrafish is necessary and sufficient for driving LPM-specific expression in all presumptive LPM progenitors from gastrulation to early somitogenesis. Panoramic SPIM and Cre/*lox*-mediated genetic lineage tracing of *drl* reporters demonstrate that the zebrafish LPM forms from a restricted mesendoderm territory during gastrulation. As upstream regulatory program read out by the *+2.0drl* pan-LPM enhancer, we identify the combination of mesendoderm transcription factors EomesA, FoxH1, and MixL1 as sufficient to drive pan-LPM activity. In cross-species assays, we observe specific activity of the zebrafish *+2.0drl* pan-LPM enhancer in LPM-corresponding territories in chicken, axolotl, lamprey, *Ciona*, and amphioxus embryos. These results demonstrate that the zebrafish *+2.0drl* enhancer reads out a universal LPM progenitor program that is conserved across chordates, defining a core transcription factor code for LPM formation. Our data provide a developmental framework for charting the earliest emergence of LPM progenitors across chordates.

## Results

**The LPM emerges as a dedicated mesendoderm population.** To resolve the dynamics of LPM emergence *in toto*, we performed time course experiments using single-plane illumination microscopy (SPIM) (Fig. 1a–d) and panoramic projections (Fig. 1e–h) of reporter-transgenic zebrafish embryos based on the full-length 6.35 kb *drl cis*-regulatory region. *drl:EGFP*-expressing LPM precursors became detectable by early gastrula stages (50% epiboly) and continuously condensed along the embryo margin through the end of gastrulation (tailbud stage) (Fig. 1a, b, f, g; Supplementary Movies 1,2). From tailbud stage onward, *drl:EGFP*-marked LPM formed a continuous band of cells with condensing anterior and posterior segments (Fig. 1c, d, g, h). We confirmed that this EGFP-positive cell band encompasses the bilateral stripes of several established LPM sub-domain markers by comparing a series of overlapping expression domains from distinct reporter lines (Fig. 1i–l). First, *lmo2:dsRED2* labels embryonic hematopoietic and vascular tissues, and its expression overlaps with medial *drl:EGFP*-expressing cells in the ALPM and PLPM (Fig. 1i). *scl:EGFP* also co-expressed with *drl:mCherry* in the most medial PLPM domain and in a small ALPM population (Fig. 1j). We find that the *pax2.1:EGFP*-expressing PLPM-derived pronephric epithelial precursors also express *drl:mCherry* (Fig. 1k). Moreover, *hand2:EGFP* expression, which demarcates the lateral-most PLPM domain plus parts of the ALPM-derived heart field and pectoral fin precursors, was also fully situated within the pan-LPM expression domain of *drl:mCherry* (Fig. 1l). Taken together, these data provide a continuous view of the emerging LPM stripes from gastrulation in zebrafish and document that the LPM emerges around the entire circumference of the zebrafish embryo (Fig. 1m).

We next sought to capture how the *drl*-expressing LPM emerges relative to the endoderm. Panoramic SPIM of the *sox17: EGFP*-positive endoderm reporter together with *drl:mCherry* revealed a population of double-positive cells from the onset of reporter detection through late gastrulation (Fig. 2a–d). After gastrulation, we detected a continuous band of *drl* reporter-positive cells around the developing embryo that was separated from the more medial endodermal *sox17* expression domain (Fig. 2d; Supplementary Movies 3,4). To confirm whether endoderm progenitors are also marked by the *drl* reporter during gastrulation, we performed *drl:creERT2*-based genetic lineage tracing with the ubiquitous *hsp70l:loxP-STOP-loxP-EGFP*

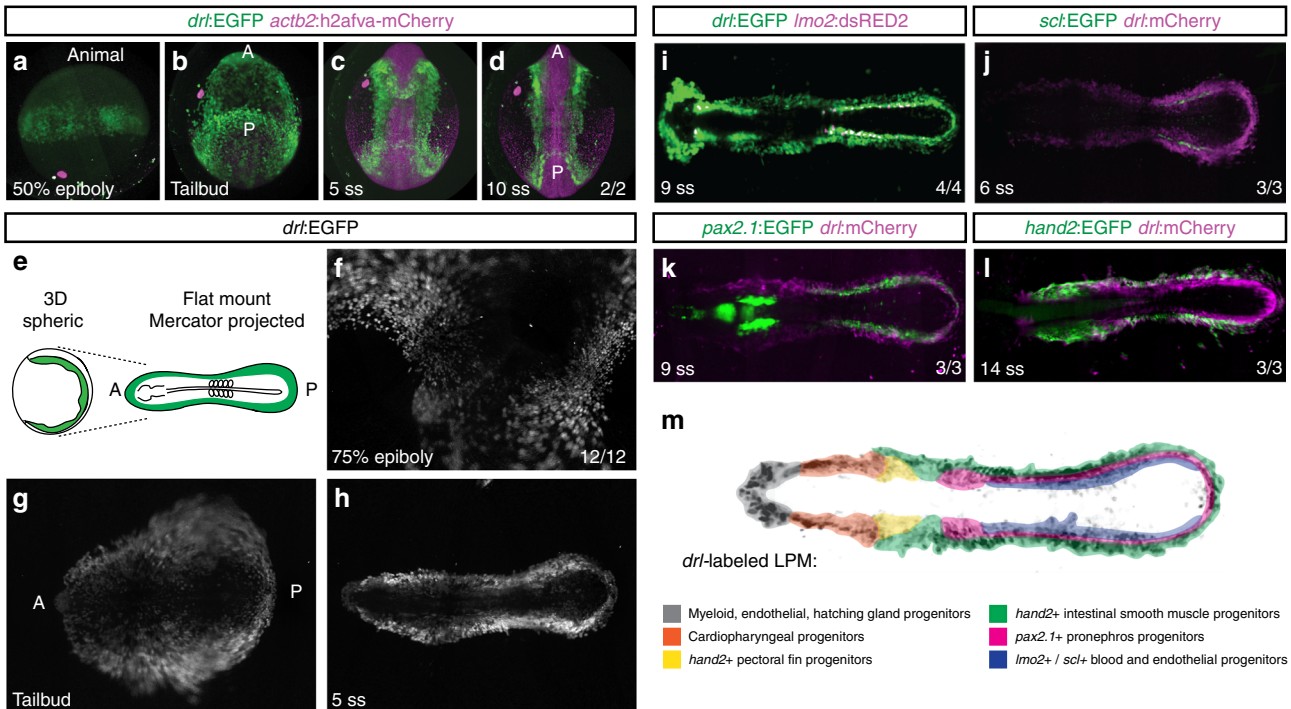

**Fig. 1** The LPM forms as a continuous field around the circumference of the developing zebrafish embryo. **a–d** Panoramic SPIM imaging of 50% epiboly to 10 ss embryos transgenic for *drl:EGFP* (green) and *actb2:h2afva-mCherry* (magenta); maximum-intensity-projected, lateral view (**a**), dorso-ventral views (anterior (A) to the top, posterior (P) bottom) (*n* = 2/2) (**b–d**). **e–h** Representative panoramic SPIM imaged *drl:EGFP* zebrafish embryo shown as 2D Mercator projection (*n* = 12/12). **e** Schematic of Mercator projection of spherical embryo, anterior to the left; **f–h** Mercator projections at 75% epiboly, tailbud, and 5 ss stages. **i–l** Single time point projections (anterior to the left) of representative double-transgenic embryos for *drl* reporters and (**i**) *lmo2* (*n* = 4/4), (**j**) *scl* (*n* = 3/3), (**k**) *pax2.1* (*n* = 3/3), or (**l**) *hand2* (*n* = 3/3) reporters co-expressed in dedicated LPM territories. **m** Summary schematic of the LPM fate territories partitioning during early somitogenesis in zebrafish

(*hsp70l:Switch*) and 4-OHT-based CreERT2 induction at discrete time points ranging from shield to 5–6 somite stages (ss) followed by analysis of labeling patterns at 72 hpf (Fig. 2e–j). 4-OHT induction at shield stage marked LPM lineages including blood, endothelium, kidney, and iSMCs as the only mesodermal fates (Fig. 2f–h)[14,15], while lineage labeling also marked broad territories within endodermal organs, including pancreas, liver, and pharynx/gut epithelium (Fig. 2h; Supplementary Fig. 1a,b). 4-OHT induction at later time points gradually decreased endoderm labeling, with minimal to no endodermal lineage signals following 4-OHT induction at 5–6 ss (Fig. 2i, j, Supplementary Figs. 1b and 2b, c). In contrast, LPM structures remained robustly labeled as the exclusive mesoderm fate, consistent with previous work[14–16]. In addition, we observed that the spatio-temporal contribution of *drl* reporter-expressing progenitors to endoderm differs along the anterior-posterior axis. We divided the embryo into four non-overlapping regions along the anterior-to-posterior axis (region I–IV) (Supplementary Fig. 2a) and quantified the switching efficiency. The amount of lineage-labeled gut endothelium increased within individual embryos from the pharynx (region I) towards the caudal gut (region IV), independent of the stage of 4-OHT administration (Supplementary Fig. 2b, c). These results indicate that progenitors expressing the *drl* reporter with ongoing development become progressively restricted to an LPM fate from anterior to posterior, until by early somitogenesis *drl* reporter expression labels only LPM.

In contrast, *sox17:creERT2* exclusively marked endoderm lineages (Supplementary Fig. 1c), supporting that *sox17* expression demarcates zebrafish endoderm progenitors downstream of the key endoderm regulator *sox32*[19]. Supporting the faithfulness

of our lineage tracing experiments, labeling does not appear skewed by any lineage-bias of the *hsp70l:Switch* reporter, as illustrated using the ubiquitous *ubi:creERT2* (Supplementary Fig. 1d). Remarkably, embryos that fail to form endoderm upon *sox32* perturbation still generated *drl*-traced LPM that partitioned into recognizable heart, blood, endothelium, and pronephros (Supplementary Fig. 3).

Altogether, these data establish that during gastrulation the *drl*-marked LPM gradually refines from a ventral-marginal mesendoderm territory to a bilateral LPM domain as the sole mesodermal fate along the entire anterior-posterior axis of the embryo. The rare lineage labeling of somitic muscle by *drl:creERT2* (Fig. 2f)[14,15] further underlines that, in zebrafish, the paraxial mesoderm and the LPM develop as distinct mesoderm lineages with only minimal overlap.

**A pan-LPM enhancer in the zebrafish *drl* locus.** The zebrafish *drl* gene encodes a putative 13-mer zinc-finger protein[20] of unknown function and is as genomic locus seemingly zebrafish-specific. First and characteristic for multimeric zinc-finger genes[21], the putative *drl* ORF lacks a clear ortholog in any other fishes or in vertebrates[17,18]. Second, *drl* is tandem-multiplied, with three *drl*-like genes featuring nearly identical ORFs and introns and a fourth more diverged putative zinc-finger gene, suggesting a recent multiplication event (Fig. 3a)[14,17]. Lastly, the *drl* locus lies in between *ap2b1* and *pex12* in the zebrafish genome, yet these two flanking genes are deeply conserved neighboring loci with no additional genes between them across vertebrates or in other fishes (Fig. 3a). Reminiscent of the also zebrafish-specific *crestin* locus that harbors a neural crest-specific

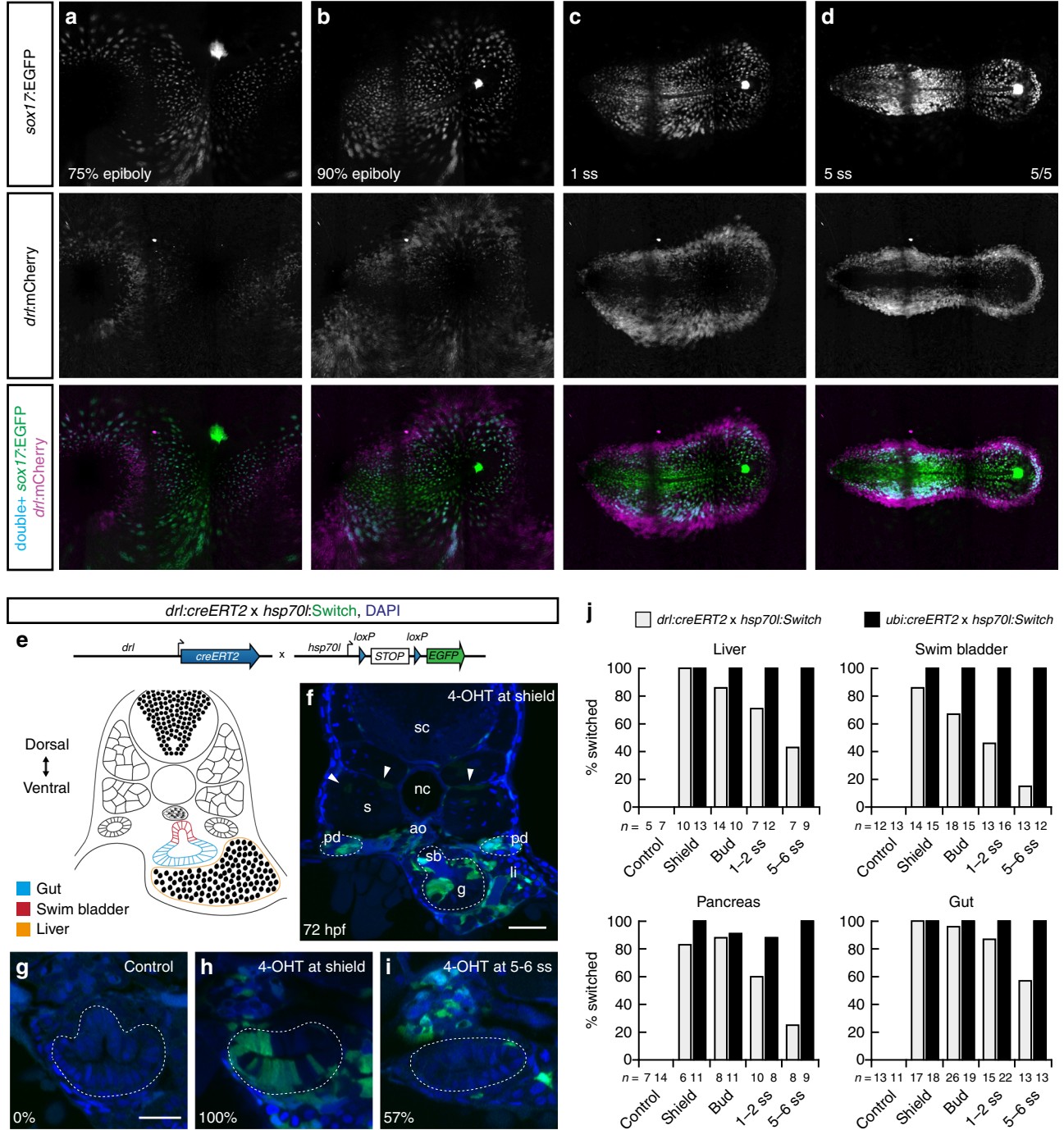

**Fig. 2** The early *drl* reporter-expressing cells comprise endoderm- and LPM-fated progenitor. **a–d** Single time point projections of *sox17:EGFP* marking endoderm progenitors (**a–d**) and *drl:mCherry* marking LPM progenitors from gastrulation until 5 ss; anterior to the left. Double-positive cells for EGFP and mCherry shown in blue (**a–d**) (*n* = 5/5). **e** Schematic of *drl:creERT2* to *hsp70l:Switch* cross for genetic lineage tracing and schematic of transverse section (trunk region) with endoderm- and LPM-derived organs at 3 dpf. **f** Representative 72 hpf transverse section of *drl* lineage-traced embryo 4-OHT-induced at shield stage; arrowheads depict rare trunk muscle labeling. **g–i** Representative transverse sections of *drl* lineage tracing at 72 hpf, control (**g**) versus 4-OHT-induced at shield stage (**h**), and 5–6 ss (**i**); note gradual loss of endoderm labeling (intestinal cells, dashed region). Numbers indicate percentage of embryos with intestinal lineage labeling. **j** Quantification of endoderm lineage labeling in representative organs following 4-OHT induction at indicated time points, comparing *drl:creERT2* versus ubiquitous *ubi:creERT2* control as reference for the *hsp70l:Switch* lineage reporter. N-numbers are mentioned in the graph legends. Notochord (nc), somite (s), spinal cord (sc), pronephric duct (pd), liver (li), swim bladder (sb), gut (g), nuclei in blue (DAPI; **f–i**). Scale bar (**f**) 50 μm and (**g–i**) 25 μm

regulatory region[22] and despite its peculiar nature, the 6.35 kb *drl* region provides a means to read out an upstream LPM input.

To identify *cis*-regulatory element(s) in the zebrafish *drl* locus responsible for pan-LPM progenitor expression, we divided the 6.35 kb *drl* regulatory region into smaller fragments and assayed their activity using Tol2-based *EGFP* reporters in F0 zebrafish and in stable transgenics (Fig. 3b). We found that the promoter-proximal region surrounding exon 1 recapitulated *drl* reporter

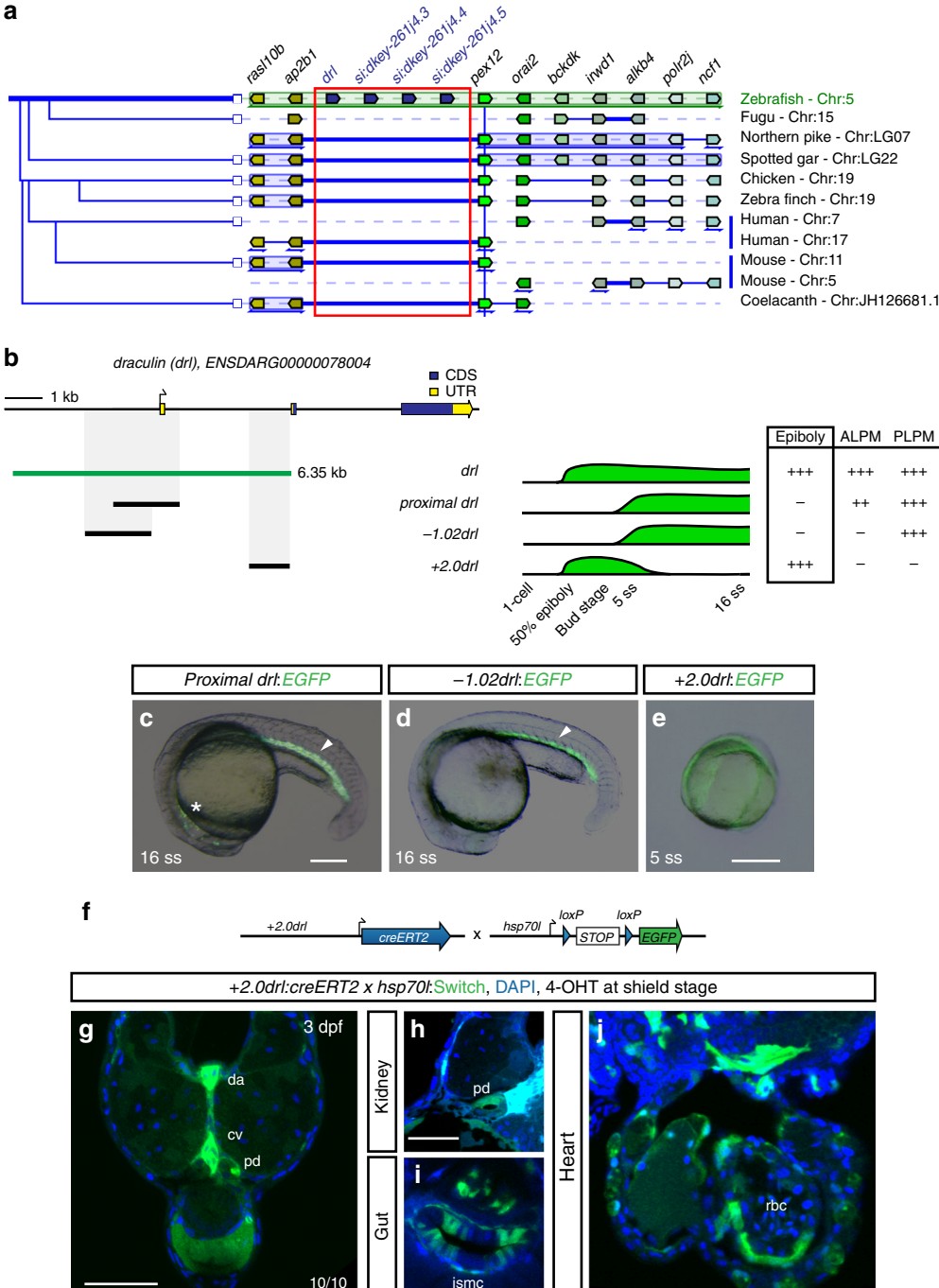

**Fig. 3** The 6.35 kb *drl cis*-regulatory region contains an early pan-LPM enhancer. **a** Adapted Genomicus-based PhyloView representation of the zebrafish *drl* locus (top row) compared to other vertebrate species. The view is centered on *pex12* (light green, blue vertical line) as anchor, with its orthologous copies across species and corresponding neighboring loci shown by their relative positions. Genes with identical coloration are homologs/orthologs. Blue arrows below gene loci indicate switched local orientation. The phylogenetic relationship tree is depicted on the left. The red box marks the *drl* locus with its neighboring three *drl*-related genes (blue) in between *ap2b1* and *pex12* in zebrafish as unique feature. Note the absence of any genes between *ap2b1* and *pex12* orthologs across fishes and other vertebrates, and conservation of suggested ancestral synteny of *ap2b1* and *pex12*. **b** Schematic of the *drl* locus depicting the 6.35 kb *cis*-regulatory region (green), and smaller isolated candidate fragments *proximal drl* (region surrounding first exon), −1.02drl (upstream region only), and +2.0drl (distal first intron) with specific reporter activity. Time line and table indicates expression dynamics (50% epiboly to 16 ss) of stable transgenes for the individual regulatory elements and expression domains (pan-LPM early or somite-stage ALPM, PLPM) with absent expression (-) to strong expression (+ +). **c–e** Representative stable transgenic zebrafish embryos harboring EGFP reporters for *proximal drl*, −1.02drl, and +2.0drl; at 5 ss and 16 ss *proximal drl* and −1.02drl express in PLPM (arrowheads in **c**, **d**), *proximal drl* additionally in ALPM (asterisk in **c**); note pan-LPM activity of +2.0drl (**e**). **f** Schematic of +2.0drl:creERT2 to hsp70l:Switch cross for genetic lineage tracing. **g–j** Transverse sections at 3 dpf of +2.0drl:creERT2 lineage tracing after 4-OHT induction at shield stage results in specific labeling of LPM-derived tissue including red blood cells (rbc), dorsal aorta (da) and cardinal vein (cv) endothelium, pronephric duct (pd), and intestinal smooth muscle cells (ismc) (*n* = 10/10). +2.0drl:creERT2 also traces endoderm-derived tissue, shown for gut epithelium. Nuclei counterstained with DAPI (blue). Scale bars (**c**, **e**, **g**) 250 μm and (**h**) 50 μm

expression in ALPM and PLPM from 5–7 ss onwards (*proximal drl*), while the promoter region alone remained active in the posterior endothelial and blood precursors (−*1.02drl*) (Fig. 3b–d). In addition, we identified a small (968 bp) region in the first intron (+*2.0drl*) that initially recapitulated early *drl* reporter expression in zebrafish embryos before fading between 5–10 ss (Fig. 3b, e). Genetic lineage tracing with +*2.0drl:creERT2* and *hsp70l:Switch* (Fig. 3f) specifically labeled LPM-derived organs including heart, blood, endothelium, kidney, pectoral fin mesenchyme, and iSMCs, and additionally marked endoderm lineages when induced with 4-OHT at shield stage (Fig. 3g–j). These results correlate well with our lineage tracing using full-length *drl:creERT2* (Fig. 2e–j)[14–16]. Further, deletion of elements within the +*2.0drl* enhancer defined a minimal enhancer region of 432 bp (+*2.4drl*) that functioned as a pan-LPM enhancer, albeit with higher variability in stable transgenics (Supplementary Fig. 4). These regulatory analyses indicate that the entire *drl* expression pattern in zebrafish derives from distinct *cis*-regulatory elements that control *drl* expression in separable early mesendoderm/pan-LPM and later ALPM versus PLPM domains. The latter pattern is analogous to the hematopoietic lineages that arise during somitogenesis and that are commonly marked with *drl* mRNA ISH[5,20]. These data imply that the +*2.0drl* enhancer contains the key regulatory modules that respond to and integrate an early LPM-defining input.

**Combined EomesA, FoxH1, and MixL1 can drive LPM formation.** We next investigated the upstream input that controls the zebrafish +*2.0drl* pan-LPM enhancer. BMP and Activin/Nodal ligands of the TGF-β superfamily trigger key pathways in early vertebrate axis determination and mesendoderm induction (Fig. 4a)[23]. During early gastrulation in zebrafish, BMP ligands are principally secreted from the ventral side, while Nodal ligands are expressed along the margin and the dorsal side[23]. In line with BMP dependence of the LPM, endogenous *drl* expression was virtually absent in embryos maternally mutant for dominant-negative Smad5 (*MZsbn*), which lack BMP activity (Fig. 4b, c)[24]. Similarly, treatment with the BMP inhibitor Dorsomorphin resulted in a pronounced decrease of endogenous *drl* expression (Fig. 4e, f). We also found decreased *drl* expression in embryos with perturbed Nodal signaling: (i) in maternal-zygotic mutant embryos lacking the key Nodal co-receptor Crypto/Oep (*MZoep*) that cannot transmit Nodal signaling around the embryo margin[25], and (ii) in embryos treated with the Nodal signaling inhibitor SB-505124 (Fig. 4d, g, h). These results indicate that the *drl* expression domain is sensitive to both BMP and Nodal input. Consistent with our lineage tracing results (Supplementary Fig. 3), embryos devoid of endoderm upon *sox32* knockdown still expressed endogenous *drl*, albeit with overall thinned-out expression and a marked decrease of dorsal *drl* activity (Fig. 4i).

We mined existing whole-embryo ChIP-seq data from zebrafish gastrulation stages[26–28] and identified candidates for transcription factors binding to the +*2.0drl* enhancer. These include the T-box transcription factor EomesoderminA (EomesA), its interaction partner FoxH1, and BMP/Nodal-mediating Smads. Published evidence has uncovered that these factors participate in controlling mesendoderm genes[29–32] and affect *drl* expression during early somitogenesis[29] (Fig. 5a). We found that mRNA injection- or *ubi* promoter-driven expression of constitutive-active forms of EomesA or FoxH1 strongly augmented and prolonged +*2.0drl* reporter and endogenous *drl* expression in their native LPM domain compared to controls (Fig. 5b–e, Supplementary Fig. 5a–d). Ubiquitous expression of wildtype *eomesA* or *foxh1* mRNA was sufficient to increase endogenous *drl* expression (Fig. 5f–h). Addition of a

constitutively-active Smad2 to EomesA and FoxH1 resulted in dorsal widening of the +*2.0drl* reporter expression pattern (Supplementary Fig. 5e, f).

EomesA and FoxH1 are maternally contributed and ubiquitously distributed during gastrulation[29,33], during which the LPM emerges in the BMP and Smad activity domain (Figs. 1,4). We therefore hypothesized that at least one additional, ventrally and marginally expressed transcription factor might be required for LPM formation. In published ChIP-seq data[26], we identified the homeodomain protein MixL1 as a third possible transcription factor that acts together with EomesA and FoxH1 in controlling +*2.0drl* enhancer activity (Fig. 5a). MixL1 is a downstream target of BMP and Nodal signaling implicated in controlling endoderm and mesoderm fates[34] and it retains ventral expression in *MZoep* mutants[29]. Furthermore, MixL1 can form a complex with EomesA[35]. Reminiscent of *eomesA* or *foxh1* mRNA injections (Fig. 5g, h), microinjected *mixl1* mRNA also resulted in increased +*2.0drl* reporter expression within in the native LPM domain (Fig. 5i).

Combining the triplet of wildtype mRNAs or Tol2-based DNA constructs encoding full-length EomesA, FoxH1, and MixL1 (shortened as *e/f/m*) led to ubiquitous +*2.0drl* reporter activation in embryos (Fig. 5j, Supplementary Fig. 5g). In *MZsbn*-mutant embryos without BMP signaling, *e/f/m* misexpression induced *drl* expression dorsally (Supplementary Fig. 5h, i). These observations suggest that in *e/f/m* overexpression conditions there is still a requirement for additional Smad activity, which in *MZsbn* embryos is only available in the dorsal Nodal-positive domain. Conversely, loss of Nodal signaling in *MZoep* mutants led to a ventral upregulation of *drl* expression upon mRNA-based *e/f/m* overexpression (Supplementary Fig. 5k, l). Combining native *e/f/m* in wildtype and *MZsbn* embryos devoid of endoderm following *sox32* knockdown also resulted in patchy, ubiquitous +*2.0drl* reporter activation (Fig. 5k, Supplementary Fig. 5j). This suggests that most, if not all, of the +*2.0drl* reporter-positive cells have an LPM identity. Mutating *mixl1* by CRISPR-Cas9 resulted in mosaic loss of +*2.0drl* reporter activity in F0 crispants (Fig. 5l–n), while mutating the *mixl1* paralog *mezzo*[36] alone or together with *mixl1* did not influence +*2.0drl* reporter activity (Fig. 5l, o, p). This indicates that MixL1 is the predominant Mix paralog acting on the +*2.0drl* enhancer in zebrafish. Furthermore, CRISPR-Cas9-mediated mutagenesis of the +*2.0drl* enhancer in the region of predicted FoxH1 and MixL1 sites in the context of the full-length *drl:EGFP* transgene resulted in specific perturbation of early LPM reporter expression, without affecting the later ALPM and PLPM patterns (Supplementary Fig. 6).

EomesA, FoxH1, and MixL1 misexpression also induced weak yet detectable expression of *tmem88a*, which is highly enriched in the early native LPM[14] (Supplementary Fig. 7a, b). In contrast, the expression domains of other early expressed mesodermal genes either showed a slight broadening of their native domains or appeared unaffected (Supplementary Fig. 7). This was illustrated further by the lack of changes in *hand2* expression, which normally initiates in the lateral-most LPM after gastrulation (Supplementary Fig. 7k–m). Together, these data suggest a regulatory model in zebrafish whereby the combination of EomesA, FoxH1, and MixL1 potentiates Smad-relayed BMP signals to demarcate a mesendoderm territory that becomes prospective LPM.

**The +*2.0drl* enhancer detects a LPM program across chordates.** We next explored whether the zebrafish-derived +*2.0drl* enhancer could read out a putative pan-LPM program in diverse chordate species. First, we revisited several previously characterized enhancers with activity in the posterior LPM of mice:

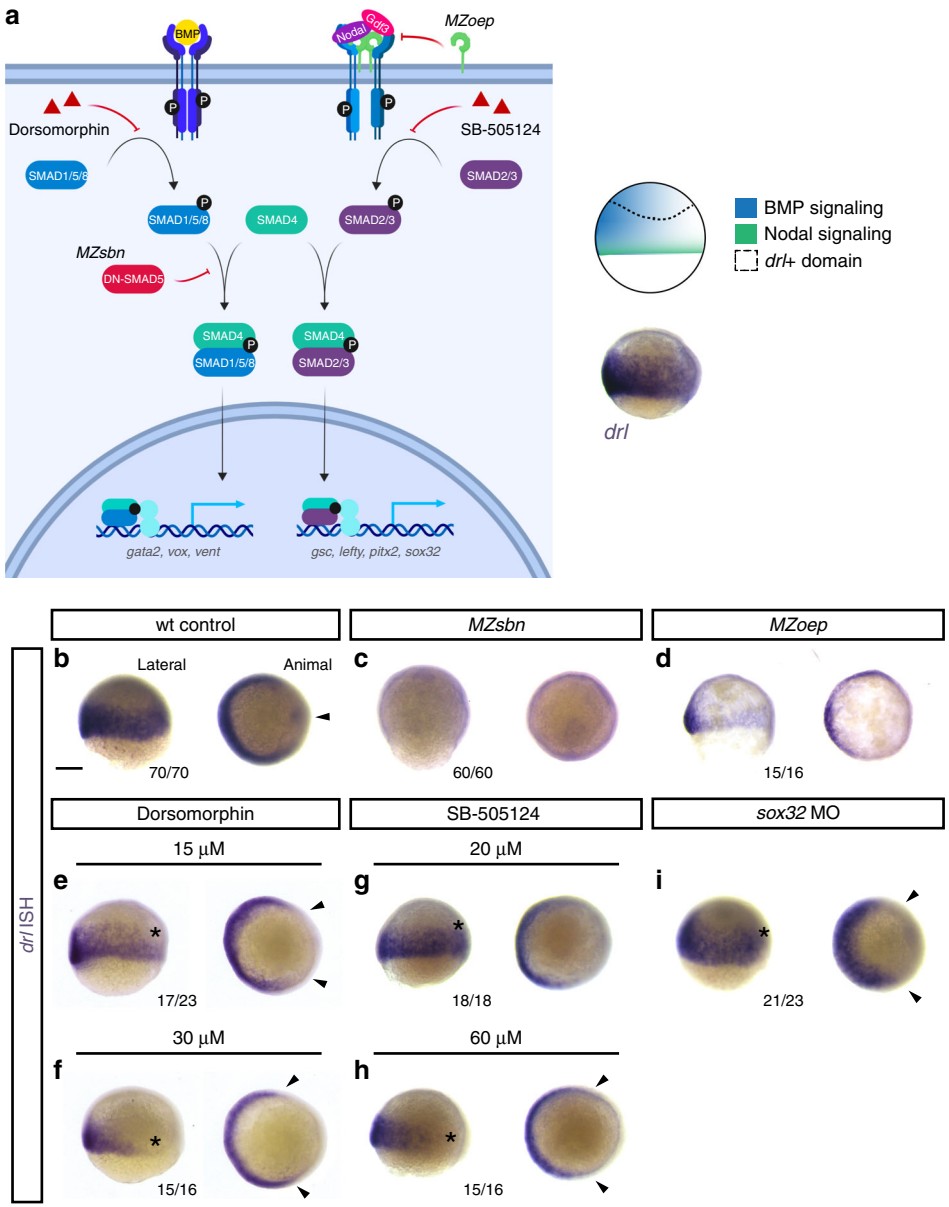

**Fig. 4** Early *drl*-expressing LPM progenitors respond to ventral BMP. **a** Schematic of the BMP and Nodal signaling pathways and the corresponding signaling territories in the gastrulation-stage zebrafish embryo, compared to endogenous *drl* expression marked by mRNA ISH. Pathway schematic generated with BioRender. **b**–**i** mRNA ISH for endogenous *drl* expression during early gastrulation (shield stage to 75% epiboly), lateral view on the left and animal view on the right. **b**–**d** Wildtype (wt) controls ($n = 70/70$), BMP-perturbed (*MZsbn*, a dominant-negative Smad5 allele, $n = 60/60$) and Nodal-perturbed (*MZoep*, mutant for the required Nodal co-receptor Oep, $n = 15/16$) showing that BMP perturbation abolishes endogenous *drl* expression. **e**–**h** Chemical inhibition of BMP via Dorsomorphin (15 µM ($n = 17/23$) and 30 µM ($n = 15/16$), **e**, **f**) decreases ventral *drl* expression, while chemical Nodal inhibition with SB-505124 (20 µM ($n = 18/18$) and 60 µM ($n = 15/16$), **g**, **h**) decreases dorsal *drl* expression, as indicated with the asterisks in the lateral views and the arrowheads in the animal views. **i** *sox32* morphants that fail to form endoderm also lose dorsal *drl* expression, as indicated with asterisk in the lateral view and with arrowheads in the animal view ($n = 21/23$). Scale bar (**b**) 250 µm

*Gata4*[12], *Bmp4*[13], and *HoxB6*[11]. Reporter transgenes based on mouse *Gata4* and *Bmp4* showed restricted activity in the outward migrating endothelial/blood progenitors and in the PLPM when electroporated into the primitive streak of *ex-ovo*-cultured chicken embryos after the onset of gastrulation (HH3+/4) (Supplementary Fig. 8a–c). The mouse *HoxB6* LPM enhancer showed no specific activity in this assay (Supplementary Fig. 8d). In contrast, when microinjected into zebrafish embryos, reporters based on these three mouse enhancers all resulted in expression mainly in the notochord without specific LPM activity (Supplementary Fig. 8e–h). This indicates that while some of the previously isolated LPM enhancers from mice express faithfully in

the PLPM of chick embryos, their activity does not recapitulate an LPM pattern in zebrafish. These results suggest that these mammalian LPM enhancers may have specialized during amniote evolution.

Electroporation of the zebrafish *+2.0drl* reporter into the primitive streak of HH3+/4 chicken embryos resulted in reporter activity specifically in the forming LPM: depending on the exact stage and region of electroporation, we observed specific reporter activity in several LPM territories. Most frequently observed expression patterns included medial and posterior LPM domains (Fig. 6a–c, Supplementary Fig. 9a–f), and we noted ALPM reaching the head fold in individual embryos (Supplementary

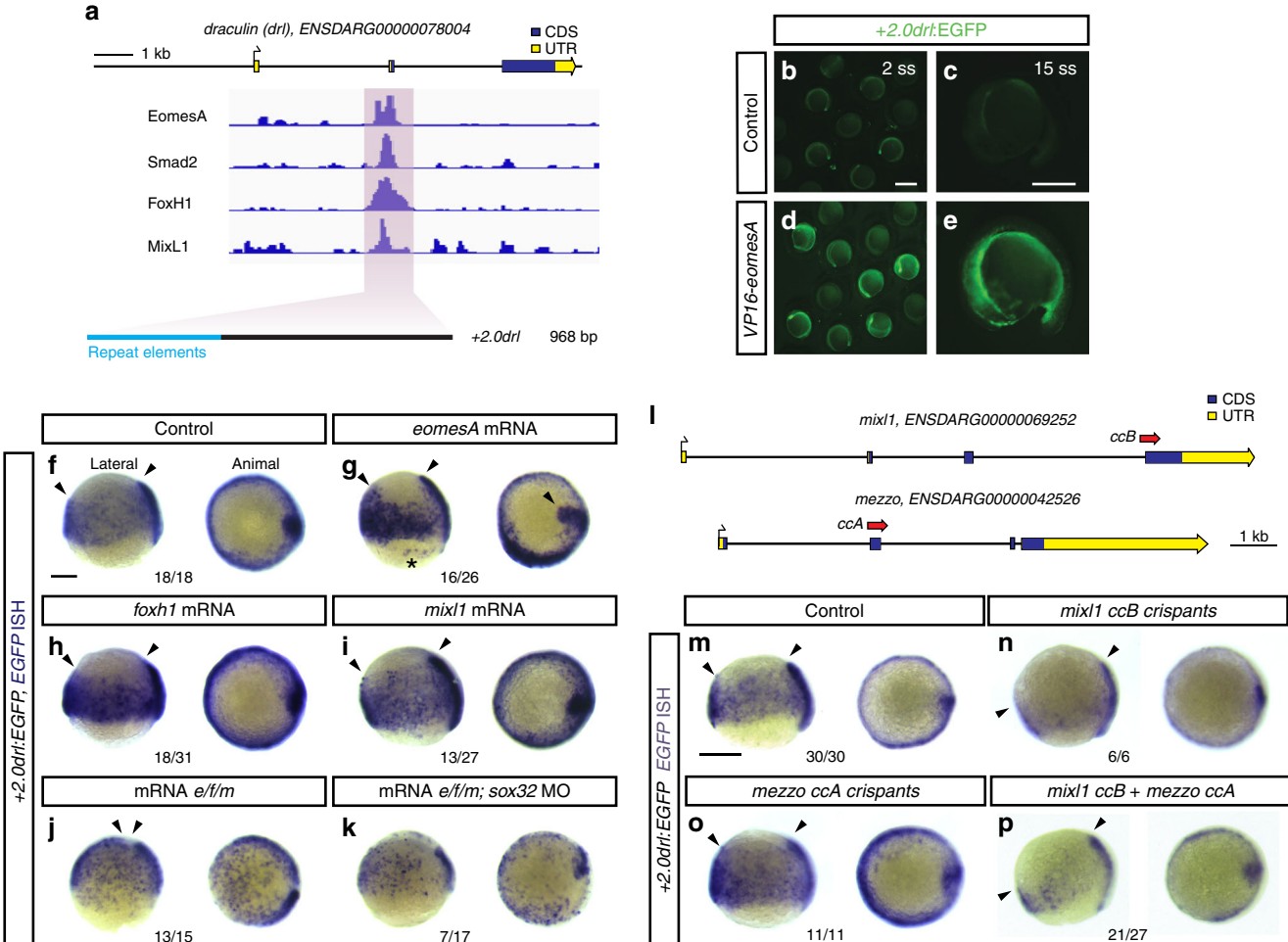

**Fig. 5** EomesA, FoxH1, and MixL1 together can induce the +2.0drl enhancer. **a** ChIP-seq tracks for EomesA. FoxH1, MixL1, and Smad2 in the drl locus. See text for details. Bottom depicts the +2.0drl intronic enhancer and the smaller, minimally specific region +2.4drl after removal of repeat sequences. **b–e** Constitutively-active VP16-EomesA boosts +2.0drl:EGFP reporter expression in its native territory. Compared to injection controls (**b**, **c**), microinjection of VP16-eomesA mRNA in +2.0drl:EGFP reporter transgenics enhances and prolongs EGFP expression in the native reporter expression domain (**d**, **e**). **f–k** Gastrulation-stage (shield to 75% epiboly) zebrafish embryos, lateral view left and animal view right, probed with EGFP ISH to detect expression of +2.0drl: EGFP. Compared to controls (n = 18/18) (**f**), embryos injected with mRNA encoding full-length eomesA (n = 16/26), foxh1 (n = 18/31), or mixl1 (n = 13/27) show enhanced +2.0drl:EGFP reporter activity (**g–i**), as indicated by arrowheads. The asterisk in **g** points out ectopic expression in the enveloping layer cells. **j** Combining mRNAs encoding full-length eomesA (e), foxh1 (f), and mixl1 (m) (e/f/m) triggers ectopic reporter expression also in dorsal blastomeres (n = 13/15, compared to native reporter expression pattern in **f**), an activity that also remained in embryos devoid of endoderm after sox32 morpholino injection (n = 7/17) (**k**). **l–p** Mixl1 acts on the +2.0drl enhancer as analyzed in Cas9 RNP-mediated crispants. **l** Schematic representation of the mixl1 and mezzo loci, with the individual sgRNAs for mutagenesis annotated (cc for CRISPR cutting, followed by sgRNA index). **m–p** mRNA in situ hybridization of EGFP expression in +2.0drl:EGFP embryos as crispant control (n = 30/30) (**m**), injected with Cas9 RNPs of (**n**) mixl1 ccB (n = 6/6), (**o**) mezzo ccA (n = 11/11), and (**p**) mixl1 ccB together with mezzo ccA sgRNA (n = 21/27); the resulting mosaic mixl1 ccB crispants show diminished +2.0drl reporter expression at late gastrulation, as pointed out by arrowheads. Lateral and animal views as indicated. Scale bar in (**b**) 500 μm, (**c**) 250 μm

Fig. 9f). These observations suggest that a basic upstream program underlying LPM formation, as read out by the +2.0drl reporter, continues to function in birds as representative amniotes.

We then tested whether zebrafish +2.0drl is responding to LPM-inducing cues in other tetrapods. Axolotl embryos microinjected with the +2.0drl EGFP reporter marked putative endodermal and LPM territories beginning from early somite-stages (st 21), additionally marking the pharyngeal regions at tailbud stages (st 27, 32; Fig. 6d–g). Transversal sections of stage 32 embryos confirmed the presence of EGFP-positive cells in the endoderm and lateral mesendoderm (Supplementary Fig. 9g–j). Notably, EGFP fluorescence was present throughout axolotl development and could be readily detected in the gut, as well as in LPM-derived tissues including the limb bud (n = 14/56; Fig. 6h,

i), heart (n = 15/56), blood vessels (n = 2/56; Fig. 6j, k), and gut lining (n = 26/56; Supplementary Fig. 9k–n). These results support the notion that the zebrafish-derived +2.0drl enhancer also interprets an LPM program active in amphibians.

Next, we asked if the +2.0drl enhancer reads out a pan-LPM program in more distantly related vertebrates. Lampreys are jawless vertebrates (cyclostomata) that can provide unique insights into vertebrate evolution due to the early divergence of their lineage from jawed vertebrates. Microinjection of the +2.0drl EGFP-based reporter into sea lamprey embryos (Petromyzon marinus) consistently resulted in robust EGFP expression in the lateral mesendoderm starting during neurulation (st 18–21) (n = 145/231; Fig. 6l, m), as well as in the developing pharynx at st 22–24 (Fig. 6n, o). Transverse embryo sections revealed that this early expression domain includes the anterior-most,

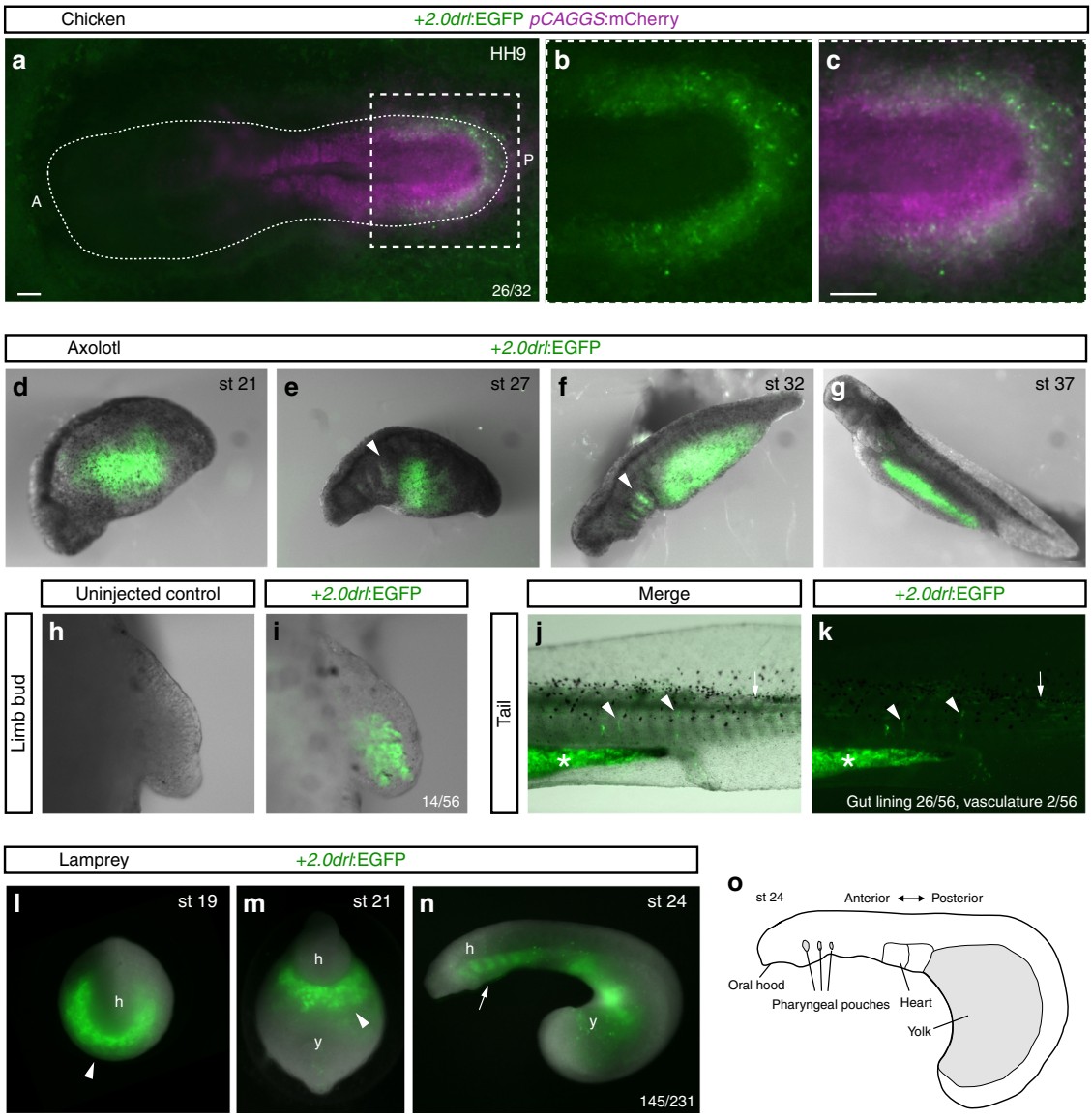

**Fig. 6** The zebrafish +2.0 drl enhancer reads out a LPM program across vertebrates. **a–c** Representative HH9 ex-ovo-cultured chicken embryo electroporated at HH3+/4 with +2.0drl:EGFP (green, **a–c**) and ubiquitous pCAGGS:mCherry control (magenta, **a**, **c**), showing specific +2.0drl reporter expression in the electroporated LPM (n = 26/32). The dashed line depicts the outline of the chicken embryo, anterior (A) to the left, posterior (P) to the right; boxed region (**a**) depicts magnified area shown for single and merged channels (**b**, **c**). **d–k** Expression of +2.0drl:EGFP upon transient injections in axolotl at the indicated embryonic stages. Note EGFP expression in the lateral portion of the embryo, future gut region, and pharyngeal arches (arrowheads in **e**, **f**). **h**, **i** EGFP expression in mesenchymal cells of the developing axolotl limb bud, indicative of LPM origin (n = 14/56). **j**, **k** EGFP fluorescence in axolotl st 43 larvae in the gut lining (asterisk, n = 26/56) and blood vessels (arrowheads, n = 2/56). Expression is also found occasionally in a small fraction of muscle fibers (arrows). **l–o** Transient transgenic lamprey embryos (Petromyzon marinus) with +2.0drl:EGFP expression in the anterior mesendoderm (arrowheads) and overlying the yolk at neurula stages (st 19–21) (**l**, **m**), and in the developing pharynx (arrow) during head protrusion (st 23–24) (**n**); views anterior (**l**), ventral (**m**), lateral (**n**), head (h) and yolk (y) (n = 145/231). **o** Schematic depiction of st 23 lamprey embryo to outline key features. Scale bar (**a**, **c**) 250 μm

LPM-linked expression of lamprey *pmHandA*[9], with the later pharyngeal expression of EGFP being restricted to the endoderm and mesoderm (Supplementary Fig. 10a–k). We conclude that the +2.0drl enhancer is capable of integrating regulatory outputs from an upstream LPM program that remains conserved across vertebrates.

Next, we asked if the +2.0drl enhancer also responds to upstream activity in LPM-linked cell fates dating back to the chordate radiation (Fig. 7a). First, we electroporated the zebrafish-derived +2.0drl reporter into embryos of the tunicate *Ciona robusta*, a chordate species belonging to a sister clade of vertebrates (Fig. 7a). While missing the full complement of LPM-

derived organ systems found in vertebrates, the LPM is echoed in the cardiopharyngeal progenitors forming in *Ciona* embryos[10,37]. We detected +2.0drl:EGFP reporter activity in emerging cardiac and pharyngeal muscle lineages at *Ciona* larval stage (st 26) (Fig. 7b–d): we observed +2.0drl:EGFP reporter activity in the atrial siphon muscle precursors (ASMPs) and in both first and second heart precursors (FHPs and SHPs). This was confirmed by co-localization of *Mesp:H2B-mCherry* expression that labels the cardiopharyngeal cell lineage (n = 15/92; Supplementary Fig. 10l, m). In agreement with the *drl*-based LPM lineage tracing in zebrafish, we found minimal to no overlap with paraxial mesoderm progenitors and the anterior tail muscles (ATMs)

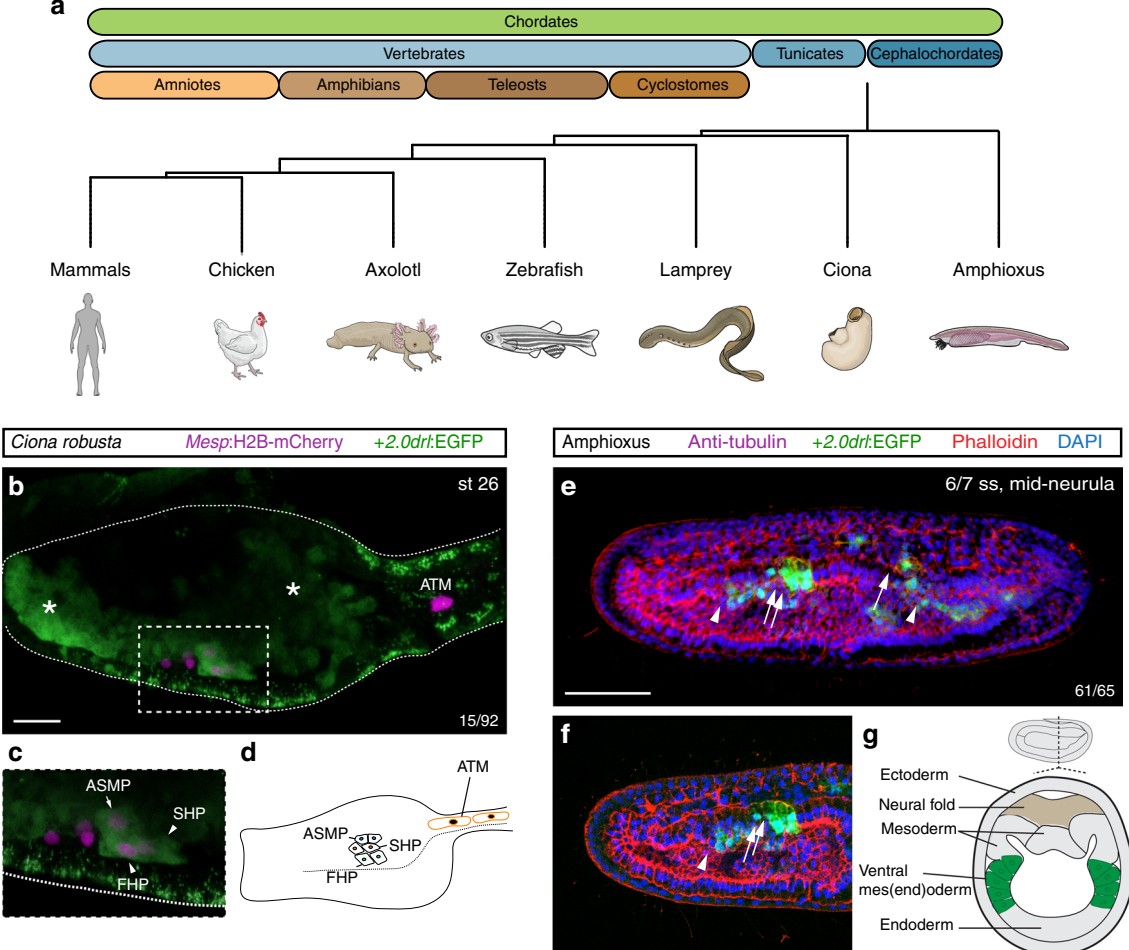

**Fig. 7** The zebrafish +2.0drl enhancer reads out an LPM program in tunicates and cephalochordates. **a** Phylogeny of chordates depicting the species used in this study. **b–d** *Ciona robusta* embryo at 18 hpf (st 26), electroporated with +2.0drl:EGFP (green), and with *Mesp*:H2B-mCherry (magenta) to track the B7.5 cardiopharyngeal cell lineage. **b**, **c** Representative larva shown with boxed region (**b**) magnified for detail (**c**); dashed line indicates midline, anterior to the left, schematic larva depicted in (**d**) (n = 15/92). +2.0drl-driven EGFP partly overlaps with B7.5 derivatives including atrial siphon muscles precursors (ASMP) (white arrows) and both first and second heart precursors (FHPs and SHPs) (white arrowheads). EGFP is also detected in mesenchymal lineages (white asterisks). **e–g** Mid-neurula stage (6/7 ss) amphioxus embryo, confocal Z-stack anterior to the left and dorsal to the top; embryo injected with +2.0drl:EGFP (green), counterstained with Phalloidin (red), DAPI (blue), lateral view as 3D rendering (**e**) and Z-stack sagittal section (**f**) (n = 61/65). +2.0drl:EGFP showing specific reporter activity in lateral mesendoderm (arrowhead), ventral half of somites (arrows), and elongating somites (double arrows). **g** Schematic depiction of amphioxus embryo; lateral view on top, dotted line represents transverse section shown below with green depicting domain of +2.0drl:EGFP expression. Scale bar (**b**) 25 μm, (**e**) 50 μm

(Fig. 7b–d). These results indicate that the zebrafish +2.0drl enhancer responds to regulatory input in the emerging multipotent cardiopharyngeal progenitors in *Ciona*.

Lastly, we examined the cephalochordate amphioxus (*Branchiostoma lanceolatum*), which belongs to the most basally divergent lineage of chordates. In amphioxus, the LPM forms from a continuous sheet of cells that encompass the dorsally emerging somites, the LPM, and the ventral-most forming endoderm[9,38–40]. At mid-neurula stage (equivalent to early somitogenesis in zebrafish), the ventral wall of the somites evaginates as nascent ventral mesoderm, which by late neurula stage fuses at the midline under the gut endoderm[9,38,39]. Indeed, the amphioxus orthologs of LPM-expressed transcription factors including *Hand*, *Csx*, *Vent1*, and *AmphiNk2-tin* are expressed in the ventral half of the somite territory at mid-neurula stage[8,38,39]. We observed that injection of zebrafish +2.0drl-based reporters into amphioxus embryos showed specific reporter activity in the ventral half of somites and in the elongating somites at mid-neurula stage (6/7 ss) (n = 61/65; Fig. 7e–g, Supplementary

Fig. 10n, o). At early larvae stage, the activity of +2.0drl-based reporter was present in the pharyngeal region (Supplementary Fig. 10p) where LPM is located[40]. Hence, also in amphioxus as cephalochordate, the +2.0drl enhancer reads out the positional input active in the LPM-corresponding territory during development.

Taken together, these observations establish that the zebrafish-derived intronic +2.0drl enhancer reads out a position-dependent LPM program that remains active in tunicates, cyclostomes, teleosts, amphibians, and amniotes, and thus across all tested chordates.

## Discussion

The dynamic nature of the LPM has made it challenging to precisely monitor its emergence and morphogenesis and has hindered comparative studies of its properties during chordate evolution. Here, we show that an enhancer from the seemingly zebrafish-specific *drl* locus (+2.0drl) reads out an

LPM-demarcating transcriptional activity in six chordate species, ranging from cephalochordates to amniotes, suggesting that this LPM-underlying transcriptional program is of ancient evolutionary origin. Characterization of the properties of this enhancer in zebrafish revealed that the transcription factors Eomes, FoxH1, and MixL1 are sufficient to trigger this basic LPM program. These observations in zebrafish suggest a regulatory model whereby, among their roles in mesendoderm regulation, Eomes, FoxH1, and Mixl1 cooperate in inducing LPM together with position-dependent Smad activity. These factors have been individually implicated in mesendoderm development in several vertebrates[27,29,32,35,41,42], in diverse LPM-associated contexts such as in blood formation[43], and in reprogramming towards cardiac and renal fates[44–46]. Based on our series of comparative and mechanistic studies, we postulate that LPM-like origins in ancestral chordates already featured the basic molecular building blocks that enabled the increasing specialization of the LPM into its sophisticated descendant cell fates observed in vertebrates.

In vertebrates, the LPM is readily detectable after gastrulation through its position lateral to the forming somites and by several progenitor markers for hemangioblast, renal, and smooth muscle fates. Genetic tracking of several aspects of LPM emergence and patterning has been achieved previously: in mouse, transgenic strains based on *HoxB6*, *Prrx1*, *Bmp4*, *Gata2*, or *FoxF1* enable labeling of the LPM post-gastrulation[11–13,47,48]. We had previously found and applied the *cis*-regulatory region of the zebrafish *drl* gene to genetically track LPM emergence during both gastrulation and early somitogenesis in zebrafish[14–16]. While *drl* encodes a putative zinc-finger protein without a clear ortholog outside of zebrafish[17,18], the early LPM-confined expression mediated by its *+2.0drl* enhancer provides a unique tool to investigate LPM origins across chordates. Using *in toto* live imaging of *drl*-based reporters together with lineage-restricted reporter transgenics, we charted LPM formation in zebrafish as a continuous process building around the entire circumference of the forming embryo (Fig. 1, Supplementary Movie 1). This mode of progenitor formation is distinct from paraxial and axial mesoderm, which both form by progressive extension over time[1].

Despite the close developmental relationship between the endoderm and the LPM, also indicated by *drl* labeling a mesendoderm population that becomes progressively dedicated towards an LPM fate (Fig. 2), LPM progenitors do not seem to require endoderm for their initial morphogenesis in zebrafish. While LPM midline migration is perturbed in *sox32* (*casanova*) mutants or morphants devoid of endoderm progenitors, such embryos still form bilateral, contracting hearts[19] and maintain kidney and iSMC progenitor markers[49]. Our LPM lineage tracing confirmed that the LPM stripes still form even without endoderm and documented how they develop into structures resembling pronephros, iSMC-like structures, endothelium, erythrocytes, and pectoral fin mesenchyme (Supplementary Fig. 3). These data imply that, despite close or even joint origin, the *sox17*-positive endoderm progenitors have minimal influence on initial LPM fate determination and LPM morphogenesis. Our imaging and lineage tracing data further indicates minimal overlap between early paraxial mesoderm progenitors and LPM progenitors, as evident in the rare occurrence of somatic muscle labeling by *drl*-expressing precursors (Fig. 2f). While there is considerable heterogeneity of cell fate domains among the post-gastrulation LPM (Fig. 1), our findings collectively suggest that the LPM initially emerges as a field of cells endowed with common properties.

Guiding the therapeutically relevant differentiation of cultured embryonic or induced pluripotent stem (iPS) cells towards cardiovascular, hematopoietic, or renal cell fates remains challenging[44,50–52]. Initial differentiation leads to broadly defined mesodermal progenitors that, depending on the protocol, show a preference to early versus late primitive streak regions, mimicking the anterior-to-posterior progression of vertebrate body axis formation[53]. Other protocols combined expression of transcription factor combinations to drive direct differentiation into specific cell fates, such as achieved for cardiomyocytes or kidney cells[2,51,54]. Nonetheless, directed differentiation of uncommitted cells into correct LPM progenitor states would be highly desirable to achieve increased reprogramming efficiency[52,55]. In this regard, our functional analyses in zebrafish showed that EomesA, FoxH1, and MixL1 together with BMP-induced Smads are able to drive cells towards an LPM program.

Eomes and FoxH1 cooperate in controlling BMP/Nodal target genes together with Smads[29,35,41]. Eomes, FoxH1, and Mixl1 have been implicated separately or in pairwise combinations in mesendoderm development[31,35,56–58]. Our findings indicate that the combination of Eomes, FoxH1, and Mixl1 modulates mesendodermal target genes required for progression towards LPM formation. The requirement for the combined action of all three factors becomes apparent when testing each factor individually, as there was only a marginal increase in *+2.0drl* expression. In contrast, the combination of all three factors was sufficient to ubiquitously induce the *+2.0drl* pan-LPM reporter. The three factors do however not merely boost mesendoderm fate per se, as demonstrated in embryos devoid of endodermal progenitors following *sox32* perturbation (Fig. 5k). The dependency of early *drl* expression on BMP, an less so on Nodal based on comparing genetic mutants, is further in line with the classic definition of a ventral-marginal emergence of the LPM[1,4]. From this data, we propose the following model: maternal Eomes and FoxH1 cooperate with BMP- and Nodal-triggered Smads to prime mesendodermal target genes. Ventral induction of MixL1 provides an instructive signal that cooperates with the previous permissive mesendoderm state to trigger an LPM fate in BMP-receiving blastomeres (Figs. 4e–i, 5f–j). These findings provide a framework for the contribution of Eomes, FoxH1, and MixL1 in programming of naïve pluripotent stem cells into cardiovascular and renal lineages by generating the correct mesendodermal precursor lineage. In-depth analysis of genomic targets of the three transcription factors is warranted to i) establish how this program conveys key LPM properties to uncommitted progenitor cells, and ii) if or which orthologs of these T-box, Forkhead, and Homeobox factors drive LPM progenitor formation across chordates.

The evolutionary origin of the LPM has remained unaddressed. In part, the discussion of evolutionary origins of key features in the vertebrate body plan is tangled by the deduction of ancestral versus derived features without an existing common chordate ancestor[59]. Jawed vertebrate species share thousands of conserved non-coding regulatory regions[60] and a greatly reduced number can be traced to jawless vertebrates like lamprey[61]. Nonetheless, while some of the previously isolated LPM enhancers of mouse *Gata4*, *Bmp4*, and *Hoxb6* expressed faithfully in the PLPM of chick embryos, their activity did not recapitulate a LPM pattern in zebrafish (Supplementary Fig. 8), suggesting specialization during amniote evolution. In contrast, our cross-species regulatory analyses of the zebrafish-derived *+2.0drl* enhancer, while as endogenous sequence absent in the tested species, uncovered a remarkable degree of regulatory conservation. While cryptic reporter activity in cross-species assays can bias results, we established that the zebrafish *+2.0drl* enhancer drove specific fluorescent reporter expression in LPM or LPM-related structures in six analyzed chordates: chick, axolotl, zebrafish, lamprey, *Ciona*, and amphioxus (Figs. 6 and 7). Of note, in *Ciona*, the *+2.0drl:EGFP* reporter also resulted in EGFP activity in cells of the developing mesenchyme; if this reporter activity is specific or ectopic activity of the used transgene plasmid as previously observed[62], remains to be determined. Nonetheless, the

zebrafish-derived +2.0drl enhancer provides a unique tool to investigate the upstream regulatory networks and the emergence of LPM structures across chordate development. This remarkable conservation of upstream reporter inputs further suggests that +2.0drl enhancer activity uncovers a deeply rooted LPM-inducing program, dating back to the last shared chordate ancestor.

Our findings further provide a genetic approach for investigating an LPM-delineating program that sets this mesodermal lineage apart from axial and paraxial mesoderm progenitors. The LPM program responds to ancient upstream regulatory inputs that defines the LPM from its early developmental origins across chordates. The activity of the +2.0drl reporter in the prospective LPM of amphioxus is particularly striking (Fig. 7), as cephalochordates form few and only rudimentary equivalents of the vertebrate LPM-derived organ systems. The LPM in amphioxus forms as a non-segmented mesodermal sheet that is continuous with the ventral prospective endoderm and the more dorsally folding somites (Fig. 7)[38–40]. This configuration makes it tempting to speculate that the LPM evolved from mesenchymal mesendoderm that did not integrate into the definitive endoderm or into the paraxial somites, providing ample material for diversification over deep time.

## Methods

**Animal experiments and husbandry.** Zebrafish and chick experiments were carried out in accordance with the recommendations of the national authorities of Switzerland (Animal Protection Ordinance). The protocols and the experiments were approved by the cantonal veterinary office of the Canton Zurich (Kantonales Veterinäramt, permit no. 150). Zebrafish care and all experimental procedures were carried out in accordance with the European Communities Council Directive (86/609/EEC), according to which all embryo experiments performed before 120 h post fertilization are not considered animal experimentation and do not require ethics approval. Adult zebrafish for breeding were kept and handled according to animal care regulation of the Kantonales Veterinäramt Zürich (TV4209). All zebrafish (Danio rerio) were raised at 25–28 °C if not indicated otherwise. White mutant (d/d) axolotls (Ambystoma mexicanum) were obtained from the axolotl facility at the TUD-CRTD Center for Regenerative Therapies Dresden, Germany. Lamprey studies were conducted in accordance with the Guide for the Care and Use of Laboratory Animals of the National Institutes of Health, and protocols were approved by the Institutional Animal Care and Use Committees of the California Institute of Technology (Protocol # 1436-11).

**Transgenic constructs and transgenic zebrafish lines.** The upstream cis-regulatory elements of the zebrafish drl gene (ENSDARG00000078004; ZDB-GENE-991213-3) were amplified from zebrafish wildtype genomic DNA and TOPO-cloned into the pENTR™ 5'-TOPO® TA Cloning® plasmid (Invitrogen) according to the manufacturer's instructions. Subsequent cloning reactions for all used transgenesis constructs were performed with the Multisite Gateway system with LR Clonase II Plus (Life Technologies) according to the manufacturer's instructions. Cloning details, transgenesis, transgenic zebrafish strains, and applied zebrafish techniques used in this study are outlined in Supplementary Methods. Primer sequences used for cloning and sgRNAs are outlined in Supplementary Table 1.

**Zebrafish transverse vibratome sections.** Fixed embryos were washed in PBS, embedded in 6% low-melting agarose (Sigma-Aldrich) in PBS/0.1% Tween-20 (Sigma-Aldrich), and cut into 130-µm-thick sections using a vibratome (Leica VT 1000 S). Sections were mounted in DAPI-containing Vectashield (Cat#H-1200; Vector Laboratories). Sections were analyzed with a Zeiss LSM710 confocal microscope with a Plan-Apochromat 40×/1.3 oil DIC M27 objective. Images were cropped and adjusted for brightness using ImageJ/Fiji[63]. Graphs were generated in GraphPad Prism 5.

**Zebrafish chemical treatments.** Chemicals for performed zebrafish treatments were dissolved in DMSO. Dorsomorphin (10–30 µM; Sigma-Aldrich) and SB-505124 (30–60 µM; Sigma-Aldrich) were administered at 1-cell stage and embryos kept in the treated E3 until fixation.

**Zebrafish selective plane illumination microscopy.** At 30–50% epiboly, embryos in the chorion were embedded into 1% low-melting agarose with optional 0.016% Ethyl 3-aminobenzoate methanesulfonate salt (Tricaine, Cat#A5040; Sigma) in E3 embryo medium, and sucked into an FEP tube (inner diameter: 2.0 mm, wall thickness: 0.5 mm). In all, 6–7 embryos were positioned on top of each other. The FEP tube was mounted in the microscope imaging chamber filled with E3 medium.

Time-lapse acquisition was performed by a standardized image acquisition pipeline[64]. The subsequent real-time image processing, registration of time points, and 2D map (Mercator) projections were performed with published Fiji scripts[64]. A Z-stack of 402 planes was obtained from every embryo with an interval of 2 min for a period of 14–17 h. Images were processed using ImageJ/Fiji and Photoshop CS6.

**Zebrafish whole-mount in situ hybridization.** Total RNA was extracted from zebrafish embryos from various stages during development. This RNA was used as template for generation of first-strand complementary DNA (cDNA) by the Superscript III First-Strand Synthesis kit (Cat#18080051; Invitrogen). In situ hybridization (ISH) probes were designed with an oligonucleotide-based method (including T7 promoter added to the reverse primers) using zebrafish cDNA (Supplementary Table 1; in situ hybridization probes). The following oligonucleotide pairs (including T7 promoter added to the reverse primers) were used to amplify the DNA template from zebrafish cDNA. The ISH probe for admp was obtained from a pCS2_ADMP plasmid, and sizzled from pCS2_Sizzled. admp and sizzled were linearized by ClaI. The gata2a probe was obtained from the middle entry vector pCM238. For in vitro transcription, T7 RNA polymerase (Roche) and digoxigenin (DIG)-labeled NTPs (Roche) were used. Afterwards, RNA was precipitated with lithium chloride, washed with 75% ethanol, and dissolved in DEPC water. RNA quality was checked on a MOPS gel. ISH on whole-mount zebrafish embryos was executed using standard zebrafish ISH protocols. After ISH, embryos were transferred to 80–95% glycerol (Sigma-Aldrich) and microscopy images were taken on a Leica M205FA with a Leica DFC450C digital camera. Images were cropped and adjusted for brightness using ImageJ/Fiji.

**Chicken embryo incubation and ex-ovo culturing.** Fertilized chicken eggs were obtained from a local hatchery and stored at 12 °C up to maximum of 14 days. Prior to use, eggs were incubated horizontally for 17 h until Hamburger-Hamilton (HH) 3 +/4 in a 39 °C incubator with 55–65% humidity. After incubation, the eggs were kept for at least 30 min at RT before opening. Eggs were opened in a petri dish and a layer of thick albumin together with the chalaziferous layer was removed using a plastic Pasteur pipette. A paper ring was placed around the embryo on the yolk and dissection scissors were used to cut the yolk membrane around the ring. The paper ring with the embryo was cleaned from remaining yolk and transferred and placed upside down on a semisolid albumin/agarose (43.5 ml thin albumin incubated for 2 h at 55 °C, 5 ml 2% agarose, 1.5 ml 10% glucose in 30 mm petri dishes) culturing plate. Embryos were recovered for at least 2 h at RT before electroporation.

**Chicken embryo injection and electroporation.** For electroporations, a customized electroporation chamber was used containing an electrode with a positive pole on the bottom of the chamber and separate negative electrode on a holder (kindly provided by the lab of Jerome Gros, Institut Pasteur, Paris). Both electrodes were mounted and connected to a square wave electroporator (BTX ECM 830). The electroporation chamber was filled with HBSS (Gibco Life Technologies), and the embryo-containing paper ring was placed in the chamber with the dorsal side up. The DNA mixture was injected by a mouth injector along the primitive streak beneath the pellucid membrane. The positive electrode holder was placed on top of the streak to allow electricity pulses flow through the embryo (3 pulses, 8 V × 50 ms, 500 ms interval). All injection mixtures for electroporations contained 0.1% fast green dye, 0.1% methyl-cellulose, 300 ng/µl control plasmid pCAGGs (pCMV: H2B-CAGG-RFP, abbreviated for chicken β-actin promoter CAGG-mCherry) and 1 µg/µl of the plasmid of interest. The embryos were placed back on the albumin culturing plates with the ventral side up and placed back at 39 °C until HH8-9. Microscopy images of the embryos were taken at HH8-9 on a Leica M205FA with a Leica DFC450C digital camera. Images were processed using Leica LAS, and cropped and adjusted for brightness using ImageJ/Fiji.

**Axolotl experiments.** The generation of transgenic animals and determination of developmental stages were performed following standardized protocols[65,66]. Animals at stage 43 were anaesthetized by bathing in 0.01% benzocaine[65]. Live imaging was performed on an Olympus SZX16 fluorescence stereomicroscope. Time lapse movies were acquired using an Axio Zoom.V16 (Zeiss) stereomicroscope. Confocal images were acquired on a Zeiss LSM780-FCS inverted microscope.

For immunostaining, embryos were fixed in MEMFA at 4 °C overnight, washed in PBS, embedded in 2% low-melting temperature agarose and sectioned by vibratome into 200-µm-thick sections. Fibronectin was detected using mouse anti-Fibronectin antibody (IST-9, mouse monoclonal, ab6328, Abcam) at 5 µg/ml.

**Lamprey experiments.** The +2.0drl regulatory element was amplified from the zebrafish vector +2.0drl:EGFP by PCR using KOD Hot Start Master Mix (Novagen) (Supplementary Table 1; regulatory elements). The amplified enhancers were cloned into the HLC vector for lamprey transgenesis[67], containing the mouse c-Fos minimal promoter, by Gibson assembly using the Gibson Assembly Master Mix (NEB).

Injections for I-SceI meganuclease-mediated lamprey transient transgenesis were performed using P. marinus embryos at the one-cell stage with injection mixtures containing 0.5 U/µl I-SceI enzyme and 20 ng/µl reporter construct. Selected EGFP-expressing embryos were fixed in MEM-FA and dehydrated in

methanol for in situ hybridization. EGFP-expressing embryos were imaged using a Zeiss SteREO Discovery V12 microscope with variable zoom and a Zeiss Axiocam MRm camera with AxioVision Rel 4.6 software. Images were cropped and adjusted for brightness using Adobe Photoshop CS5.1.

For mRNA ISH, total RNA was extracted from st 21–26 *P. marinus* embryos using the RNAqueous Total RNA Isolation Kit (Ambion). This was used as a template for 3′ rapid amplification of cDNA ends (RACE) with the GeneRacer Kit and SuperScript III RT (Invitrogen). A 339bp-long *pmHandA* in situ probe was designed based on a characterized cDNA sequence from the closely related Arctic lamprey (*Lethenteron camtschaticum*)[68], and this sequence was amplified by PCR from 3′ RACE cDNA using KOD Hot Start Master Mix (Novagen) with the following primers: 5′-GCGGAGGACATTGAGCATC-3′ (forward) and 5′-TGGAATTCGAGTGCCCACA-3′ (reverse). The cDNA fragment was cloned into the pCR4-TOPO vector (Invitrogen).

Lamprey whole-mount ISH was performed using DIG-labeled probes including for *eGFP*[67]. Embryos were cleared in 75% glycerol and imaged using a Leica MZ APO microscope with variable zoom and Lumenera Infinity 3 camera with Lumenera Infinity Capture v6.5.3 software. Images were cropped and adjusted for brightness using Adobe Photoshop CS5.1.

After ISH, selected embryos were transferred into 30% sucrose in PBS, embedded in O.C.T. Compound (Tissue-Tek), and cut into 10 μm-thick cryosections using a CryoStar NX70 cryostat (Thermo Scientific). Images were taken using a Zeiss Axiovert 200 microscope with an AxioCam HRc camera and AxioVision Rel 4.8.2 software.

***Ciona* experiments**. *+2.0drl* was amplified from the zebrafish vector *+2.0drl:EGFP* and sub-cloned upstream of *unc76:GFP* to generate a *Ciona* reporter construct including minimal promoter (*pBuS24*; see Supplementary Table 1, regulatory elements for primer sequences). Gravid *Ciona robusta* adults were obtained from M-REP (San Diego CA, USA). To test the activity of the zebrafish enhancers in *Ciona robusta*, 80 μg of *+2.0drl:EGFP* was injected in a mixture with the reporter plasmid for *Mesp*[69] to mark the B7.5 cardiopharyngeal lineage with *H2B:mCherry* (10 μg). For antibody staining, embryos were fixed in 4% MEM-PFA for 30 min, rinsed several times in PBT (PBS/0.1% Tween-20), and incubated with anti-GFP (1:500, mouse mAb, Roche) with 2% normal goat serum in PBT at 4 °C overnight. Embryos were washed in PBT and then incubated with donkey anti-mouse secondary antibody (1:1000) coupled to Alexa Fluor 488 (Life Technologies) in PBT with 2% normal goat serum for 2 h at RT, then washed in PBT[70].

**Amphioxus experiments**. The regulatory elements *drl* (entire 6.35 kb) and *+2.0drl* were amplified from the zebrafish reporter vector *drl:EGFP*[14] and *+2.0drl:EGFP* and subcloned upstream of a EGFP reporter in the *pPB* vector carrying *PiggyBac* transposon terminal repeats[71]. Adults of *Branchiostoma lanceolatum* were collected in Banyuls-sur-Mer, France, prior to the summer breeding season and raised in the laboratory until spawning. The spawning of amphioxus male and females was induced by shifting of the temperature[72]. For microinjection of amphioxus eggs, a mixture of *pPB-drl:EGFP* or *pPB- +2.0drl:EGFP* (200 ng/μl) with PiggyBac transposase mRNA (100 ng/μl) in 15% glycerol was used. Transgenic embryos were allowed to develop until neurula stage, fixed in 4% PFA overnight at 4 °C, stained with monoclonal anti-acetylated Tubulin antibody (T6793 Sigma-Aldrich, mouse ascites fluid, clone 6-11B-1, dilution 1:500), mounted with Vectashield with DAPI (Vector Laboratories), and analyzed using a Leica SP5 confocal microscope. The confocal images were adjusted for brightness and contrast with ImageJ/Fiji.

**Reporting summary**. Further information on research design is available in the Nature Research Reporting Summary linked to this article.

## Data availability
The authors declare that the data supporting the findings of this study are available within the paper and its supplementary information files. Original data underlying the lamprey experiments in this manuscript are accessible from the Stowers Original Data Repository at http://odr.stowers.org/websimr/.

The source data underlying Fig. 2j and Supplementary Fig. 2b and 10m are provided as a Source Data file. Reagents are available upon request.

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

## Acknowledgements

We thank Seraina Bötschi, Lukas Obernosterer, and Vesna Barros for technical and husbandry support; the lab of Stephan Neuhauss for zebrafish husbandry support; the labs of Esther Stöckli and Jerome Gros for chicken experimentation support; the ZMB at UZH for imaging support; Fiona Wardle for input and support on the ChIP-seq panel in Fig. 5a; the lab of Magdalini Polymenidou for vibratome access; Karolína Ditrychová for cloning the *pKD001* construct; Rebecca Burdine for sharing transcription factor constructs; Mark Miller for animal illustrations in Fig. 7a; the Stowers Institute histology facility for assistance with lamprey embryo sectioning; Hans-Henning Epperlein for discussions on salamander embryology; Diego Safian for suggesting Genomicus for synteny analysis; and all members of the Mosimann lab for constructive input.

This work has been supported by a Swiss National Science Foundation (SNSF) professorship [PP00P3_139093] and SNSF R'Equip grant 150838 (Lightsheet Fluorescence Microscopy), a Marie Curie Career Integration Grant from the European Commission [CIG PCIG14-GA-2013-631984], the Canton of Zürich, the UZH Foundation for Research in Science and the Humanities, the Swiss Heart Foundation, and the ZUNIV FAN/UZH Alumni to C.M.; a UZH CanDoc to C.H.; EuFishBioMed and Company of Biologists travel fellowships to K.D.P.; the Stowers Institute (grant #1001) to H.J.P. and R.K.; NIH/NHLBI R01 award HL108643, trans-Atlantic network of excellence award 15CVD01 from the Leducq Foundation to L.C.; a long-term fellowship ALTF 1608-2014 from EMBO to C.R.; Alexander von Humboldt fellowship to A.C. and DFG Research Center (DFG FZ 111) and Cluster of Excellence (DFG EXC 168) funds to M.H.Y.; Czech Science Foundation 17-15374 S to Z.K.

## Author contributions

K.D.P., C.H., S.N., E.C.B., S.B., and C.M. designed, performed, and interpreted the zebrafish experiments; S.N., E.C., and A.B. established and performed the chicken experiments; K.D.P. performed the lightsheet imaging with technical and equipment support by G.S. and J.H.; K.W.R. and P.M. provided and generated mutants and maternal-zygotic mutant zebrafish; H.J.P., M.B., and R.K. designed, performed, and interpreted the lamprey experiments; A.C., D.K., and M.H.Y. designed, performed, and interpreted the axolotl experiments; C.R. and L.C. designed, performed, and interpreted the *Ciona* experiments; I.K. and Z.K. designed, performed, and interpreted the amphioxus experiments; A.C., D.K., and M.H.Y. designed and performed the axolotl experiments; K.D.P., C.H., S.N., and C.M. assembled and wrote the manuscript with contributions from all co-authors.

## Additional information

**Competing interests:** The authors declare no competing interests.

