## [Peer Review File · Nature Communications]

Reviewers' Comments:

Reviewer #1:

Remarks to the Author:

In this manuscript, Mosimann and his co-workers isolated the zebrafish *drl* enhancer that drives its reporter expression throughout the lateral plate mesoderm (LPM) from early gastrulation. They also identified the transcription factors driving the LPM specific *drl* reporter activity. In addition, they examined the zebrafish *drl* reporter activity in diverse chordate species.

Main concerns

line 283, "*drl* encodes a putative zing-finger protein of unknown function without a clear ortholog outside of zebrafish". It is impossible to discuss the evolutionary interpretation and implication of the zebrafish *drl* enhancer sequence without knowing the orthologous genes or their neighboring genomic sequences in other chordate species.

I am afraid that authors are not able to show the following necessary requirements in this work:

The expression pattern of *drl* orthologs in mouse, chicken, axolotl, lamprey, *Ciona* and amphioxus are required.

The enhancer sequences of *drl* orthologs in other chordate species, as well as the putative binding region of the identified TFs in each enhancer, should be shown.

The function of the enhancer sequences of *drl* orthologs in other chordate species should be examined.

Minor points:

1. Fig.2E; the color of "gut" and "swim bladder" are too similar.
2. lines 815, 816; 'h' or 'rbc' are not in Fig. 3.
4. Fig. 4 E –I, Fig. 5F-K, M-P; Differences in expression levels/patterns are not clear in some cases. It is required to show quantitatively that is an increase or decrease in *drl* expression levels.
5. Fig. 1, 2, 3F, 5B-E, 6; sample numbers are required.
6. Fig. 1, 2, 3A, 5B-E, 6, 7E, F, S2, S3, S8, S9chicken, S10A-H, N-P; scale bars are required.

Reviewer #2:

Remarks to the Author:

This is an interesting study – the authors uncovered a Lateral Plate Mesoderm (LPM) enhancer for the gene *drl* – and use it to define the progenitor field and show it's conserved in many chordates.

The implication is that this might represent an ancient lineage (LPM).

Using lineage tracing in Zebrafish, they show that the LPM and the paraxial mesoderm develop as distinct mesodermal lineages. Using microinjection, they then show that the 2kb-*drl* enhancer also expresses in LPM in axolotl, lamprey, while in *Ciona* it expressed in the heart and pharyngeal muscle.

This enhancer therefore seems to incorporate input from the same transcription factors and represent the ancient code for LPM development.

Overall, it's a very well conducted study, and a nicely written paper and should be published as is. I have to say, in all my time reviewing, this is the first time I have ever recommended this - so I congratulate the authors for a study well done.

Reviewers' comments:

Reviewer #1 (Remarks to the Author):

In this manuscript, Mosimann and his co-workers isolated the zebrafish *drl* enhancer that drives its reporter expression throughout the lateral plate mesoderm (LPM) from early gastrulation. They also identified the transcription factors driving the LPM specific *drl* reporter activity. In addition, they examined the zebrafish *drl* reporter activity in diverse chordate species.

Main concerns

line 283, “*drl* encodes a putative zing-finger protein of unknown function without a clear ortholog outside of zebrafish”. It is impossible to discuss the evolutionary interpretation and implication of the zebrafish *drl* enhancer sequence without knowing the orthogous genes or their neighboring genomic sequences in other chordate species.

I am afraid that authors are not able to show the following necessary requirements in this work:

The expression pattern of *drl* orthologs in mouse, chicken, axolotl, lamprey, ciona and amphioxus are required.

The enhancer sequences of *drl* orthologs in other chordate species, as well as the putative binding region of the identified TFs in each enhancer, should be shown.

The function of the enhancer sequences of *drl* orthologs in other chordate species should be examined.

We appreciate the reviewer's input and constructive criticism of our work. We do believe that the reviewer might have misunderstood a small, yet significant detail of our manuscript's starting point. This misunderstanding likely stems from our too extensive trimming of a more detailed introduction on the nature of the *draculin* (*drl*) gene.

The *drl* locus is zebrafish-specific and does not feature any clearly assignable ortholog in other vertebrates, not even in closely related fishes. While known in the zebrafish hematopoiesis community, this is also documented in the literature and in our previous work by several lines of evidence:

1) *drl* encodes a putative 13-mer zinc finger protein, a protein family that evolves so rapidly that even between great apes and humans orthologs are difficult to assign (Emerson & Thomas, 2009). Specifically for *drl*, BLAST does pick up hits to other members of the vast zinc finger transcription factor family, but matches are restricted to the zinc finger domains and cannot be used to assign a clear ortholog in any species outside of zebrafish (Pimpong et al., 2014; Sumanas et al., 2005).

2) *drl* is located in between the *ap2b1* and *pex12* genes in the zebrafish genome. Genome comparisons indicate that *ap2b1* and *pex12* are ancestral neighboring genes, with no other loci in-between them. Yet strikingly, and different to other fishes, only zebrafish have the *drl* locus inserted between these two ancient

neighbors. The hypothesis is that *drl* has transposed into this locus in the zebrafish lineage and has rapidly evolved (as multimeric zinc finger genes do).

3) The zebrafish *drl* locus altogether consists of four zinc finger genes, of which three are almost identical copies of *drl* and the fourth is again a highly similar zinc finger gene (neighboring *pex12*) in zebrafish. We had previously shown that these zinc finger genes are flanked by zebrafish-specific transposons of the DANA family and repetitive stretches (Mosimann et al., 2015). Together with the highly similar ORFs and even non-coding introns among the *drl* loci (Pimtong et al., 2014), this all hints at a very recent gene duplication/multiplication event after transposition in-between *ap2b1* and *pex12* from an unknown original location in the zebrafish genome.

Therefore, the major points raised by Reviewer 1 suggest that we need to introduce this better for the reader. We have now added a paragraph introducing the context of the *drl* locus and added Supplementary Figure 3 to document the synteny of *ap2b1*, *pex12*, and the unique insertion of the *drl* locus in zebrafish.

Of note, we had previously published work on the unrelated, yet also uniquely zebrafish-specific *crestin* gene in neural crest and melanoma (Kaufman et al., 2016). *drl* now further underlining the utility of species-specific regulatory elements to decipher conserved developmental progenitor states across species.

Minor points:

1. Fig.2E; the color of “gut” and “swim bladder” are too similar.

We appreciate the reviewer’s eye – now corrected in the figure.

2. lines 815, 816; ‘h’ or ‘rbc’ are not in Fig. 3.

Resolved.

3. Fig. 4 E –I, Fig. 5F-K, M-P; Differences in expression levels/patterns are not clear in some cases. It is required to show quantitatively that is an increase or decrease in *drl* expression levels.

The used colorimetric mRNA in situ derives its color intensity based on incubation time, temperature, and probe concentration, all variables that can vastly diverge between experiments, labs, and years. The intent of the used method is therefore rather the documentation of qualitative differences. We now clarify the qualitative changes using arrow heads and asterisks how and where *drl* or *GFP* reporter expression changed, as customary in the field.

4. Fig. 1, 2, 3F, 5B-E, 6; sample numbers are required.

We have now added the sample numbers with the individual experiments, and double-checked to have these presented for all experiments throughout the manuscript. For Figure 5B-E, the experiment depicted was the qualitative observation that led to the quantification shown in Figure 5F,G.

5. Fig. 1, 2, 3A, 5B-E, 6, 7E, F, S2, S3, S8, S9chicken, S10A-H, N-P; scale bars are required.

We have revisited scale bars for consistency throughout the manuscript. For the Mercator projections in Fig. 1 and 2 scale bars are not possible as the projections are distance-distorted and do not allow homogeneous assessment of distances (Schmid et al., 2013). Lamprey (Fig. 6 and SFig. 10) and axolotl (Fig. 6) images were not taken using software-coded microscopes that would allow size estimates. Scale bars would be estimated and thus imprecise; we therefore prefer the current status.

Reviewer #2 (Remarks to the Author):

This is an interesting study – the authors uncovered a Lateral Plate Mesoderm (LPM) enhancer for the gene *drl* – and use it to define the progenitor field and show it's conserved in many chordates.

The implication is that this might represent an ancient lineage (LPM).

Using lineage tracing in Zebrafish, they show that the LPM and the paraxial mesoderm develop as distinct mesodermal lineages. Using microinjection, they then show that the 2kb-*drl* enhancer also expresses in LPM in axolotl, lamprey, while in *Ciona* it expressed in the heart and pharyngeal muscle.

This enhancer therefore seems to incorporate input from the same transcription factors and represent the ancient code for LPM development.

Overall, it's a very well conducted study, and a nicely written paper and should be published as is. I have to say, in all my time reviewing, this is the first time I have ever recommended this - so I congratulate the authors for a study well done.

We much appreciate the reviewer's take on our manuscript, and are humbled by her/his assessment. We hope our work will provide a fruitful platform to expand our understanding of LPM biology across models.

Reviewers' Comments:

Reviewer #1:

Remarks to the Author:

I do understand that there is no ortholog of *drl* outside of zebrafish, and thus authors cannot identify the function of orthologous genes. If the lateral plate mesoderm is specific to zebrafish, we would find the biological significance in this zebrafish enhancer. However, we would not be able to discuss the generality or the evolutionary origin of the lateral plate mesoderm, which is formed in all vertebrates, based on the function of the enhancer specific to zebrafish.

Thus, I recommend that this work should be rejected for publication.

Reviewers' Comments:

Reviewer #1:

Remarks to the Author:

I do understand that there is no ortholog of *drl* outside of zebrafish, and thus authors cannot identify the function of orthologous genes. If the lateral plate mesoderm is specific to zebrafish, we would find the biological significance in this zebrafish enhancer. However, we would not be able to discuss the generality or the evolutionary origin of the lateral plate mesoderm, which is formed in all vertebrates, based on the function of the enhancer specific to zebrafish.

Thus, I recommend that this work should be rejected for publication.

Response:

In our work, we use the intronic 2.0*drl* enhancer as sensor for a regulatory input that is common to LPM progenitor cells. While we appreciate that individual gene function across species can vary, cis-regulatory elements that respond to upstream input across species do so disconnected from the function of the gene controlled by the cis-regulatory element.

Cis-regulatory elements that do not necessarily show conservation in other species have been previously harnessed to successfully assign evolutionary connections between regulatory programs (i.e. PMID 16556802, 30867425). Further, seemingly species-specific cis-regulatory elements have provided valuable insights into conserved regulatory programs in development and disease (PMID 22011226, 26823433, 30464347). The +2.0*drl* enhancer now provides a first opportunity to probe LPM emergence across chordates. We hope our work will provide new impulses for the field and will fuel the quest for conserved LPM enhancers across vertebrates or even chordates to decipher the regulatory logic forming this fascinating mesoderm territory.